# Transferring SLU Models in Novel Domains

## Abstract

Spoken language understanding (SLU) is a critical component in building dialogue systems. When building models for novel natural language domains, a major challenge is the lack of data in the new domains, no matter whether the data is annotated or not. Recognizing and annotating "intent" and "slot" of natural languages is a time-consuming process. Therefore, spoken language understanding in low resource domains remains a crucial problem to address. In this paper, we address this problem by proposing a transfer-learning method, whereby a SLU model is transferred to a novel but data-poor domain via a deep neural network framework. We also introduce meta-learning in our work to bridge the semantic relations between seen and unseen data, allowing new intents to be recognized and new slots to be filled with much lower new training effort. We show the performance improvement with extensive experimental results for spoken language understanding in low resource domains. We show that our method can also handle novel intent recognition and slot-filling tasks. Our methodology provides a feasible solution for alleviating data shortages in spoken language understanding.

## 1 Introduction

The recent surge of artificial intelligence motivates the technical and applicable exploration of novel human-computer interactions. Spoken dialogue systems are widely studied and used in various mobile devices. Well-known commercial applications driven by dialogue systems include intelligent personal assistants and intelligent robots for customer services. As more and more products and scenarios integrate dialogue systems in intelligent services, the ability for model adaptation in dialogue systems is critically needed. Many state-of-the art dialogue systems follow a learning pipeline that includes components such as spoken language understanding (SLU), dialogue management (DM) and spoken language generation or retrieval (SLG). Since both dialogue management and spoken language generation systems rely on knowledge or information learned from spoken language understanding, research and industry community have paid much attention to the SLU area. In this paper, we focus on the problem of data shortage for SLU in a new domain.

Spoken language understanding typically defines and represents user utterances in terms of semantic frames comprised of domains, intents and slots (Tur & Mori, 2011). A spoken language understanding task involves classifying the domain, detecting the intent of user utterance and identifying token sequences corresponding to slot values in the semantic frame. A learning model is often designed to handle the above sub-tasks separately and sometimes simultaneously. Deep learning models are the cutting-edge techniques for achieving impressive performance in spoken language understanding compared with conventional machine learning models.

However, deep neural-network models often require the collection and manual annotation of data, which is a time and labor intensive process. These problems pose a major challenge to building high-quality learning models in new domains. To address this problem, we propose to perform domain adaption of SLU models by transferring the SLU knowledge and model parameters learned in an auxiliary domain where a model is already built. A related problem is that only few examples of some output intent and slot classes are available during the training phrase in a new domain, because training sets for a new domain are small. To deal with this limitation, we propose a novel few shot learning model in a meta learning paradigm, we call it the Transfer Semantic Similarity Model (TSSM). The TSSM achieves few-shot learning by exploiting the semantic embeddings of both seen and unseen intent and slots.

In summary, the challenges of spoken language understanding include: (1) Expanding SLUs to novel domains is necessary as the development of artificial intelligent products extends to more areas in our society. But it is difficult to get sufficient data or spoken language sentence in novel domains. (2) Without sufficient labeled data, deep learning methods are difficult to train and thus it is hard to obtain satisfactory performance in a new domain. (3) Unseen intents and slots exist because small annotated training set might not cover all intents and slots. Thus, the generalization ability of the models is crucial for dealing with unseen classes.

Our contributions are summarized as follows:

- We design a novel and efficient SLU model, the TSSM. It incorporates transfer learning and meta learning for SLU tasks in low resource domains.

- The proposed TSSM can improve model performance with only a small amount of training data in new domains and it can handle unseen classes. The experiments illustrate that our proposed methods outperform the state-of-the-art baselines in many experimental domains.

- Our work can treat intent and slot detection tasks as a structured multi-objective optimization problem, which further improves the accuracy of learning tasks in new domains.

## 2 RELATED WORK

Spoken language understanding is usually defined as a supervised learning problem, involving conventional machine learning models (Young, 2002; Hahn et al., 2011; Wang et al., 2005) and deep learning models (Mesnil et al., 2015; Kurata et al., 2016; Sarikaya et al., 2011) on massive amount of annotated training data. Although unsupervised learning and semi-supervised learning based approaches have been proposed as well (Chen et al., 2013; 2015b;a), deep learning approaches were shown to outperform most others. Besides the requirement of a large amount of annotated data being available, the intents and slots are usually predefined. As a result, they are inflexible to expand to new domains. Some researchers have paid attention to alleviate this limitation. An example is the work by Korpusik et al. , which retrains models to cover new intents and associated slots while redesigning a semantic schema (Korpusik et al., 2014).

There have also been some attempts to learn shared latent semantic representations or model parameters for multiple domains via transfer learning. A survey for transfer learning with applications is Pan & Yang (2010). In general, the goal of transfer learning is to improve performance in novel domains that only has small data. In these target domains, it is critical to reduce the model re-training efforts by bridging the knowledge from source tasks to the new target tasks. In natural language processing (NLP), researchers have studied cross-domain NLP problems but have mainly focused in cross-lingual problem settings (Mou et al., 2016; Buys & Botha, 2016; Kim et al., 2017a;b). In Hakkani-Tr et al. (2016), both multi-task and transfer learning approaches are proposed to learn shared implicit feature representation across various domains for spoken language understanding. Researchers in Amazon have shown in a recent study that DNN-based natural language engines can re-use the existing models through transfer learning (Goyal et al., 2018).

When the training data is missing some task labels, one can exploit meta learning to apply few-shot learning to help with the knowledge transfer Fei-Fei et al. (2006); Palatucci et al. (2009); Vinyals et al. (2016); Ravi & Larochelle (2018). For example, Vinyals et al. (2016) learns a network that maps a small amount of labeled support set and unlabeled examples to its labels, saving the effort in fine-tuning to new class types. Besides image object-recognition tasks, in SLU some experiments have been conducted. The objective of Ferreira et al. (2015) is to predict the semantic tag sequences of a user query without using any target-domain user utterances and thus in-context semantic tags. The research builds a statistical model to predict the SLU for unseen data in Yazdani & Henderson (2015); Chen et al. (2016). An action-matching algorithm is proposed by Zhao & Eskénazi (2018) to learn a cross-domain embedding space that models the semantics of dialog responses, which, in turn, lets a neural dialog generation model generalize to new domains. Recent publication by Google (Bapna et al., 2017) explores semi-supervised slot-filling based on deep learning that can use the slot labels without the need for any labeled or unlabeled data in domain examples or the need for any explicit schema alignment, to quickly bootstrap the model in a new domain.

## 3    OUR PROPOSED APPROACH FOR MODEL TRANSFER

In this section, we introduce our transfer learning approach to address the cross-domain SLU problem. We consider a source domain $D_s$ and a target domain $D_t$, where our goal is to learn a model in the target domain. Typically, we have a large annotated dataset for $D_s$, while only a few labeled data for $D_t$. The tasks in both $D_s$ and $D_t$ are of the same types, namely they are both slot filling and intent detection tasks. However, $D_s$ and $D_t$ have different intent label spaces $\mathbb{T}_s$ and $\mathbb{T}_t$, because the target domain might have new intent values (slot faces the same situation).

The number of possible intents and slots grows rapidly as we get into new domains with new intents and slots. One natural approach in solving the learning problem is to train one binary classifier for each possible label, and decide whether or not to include the classification model in the output. However, this would require training a large number of classifiers. It also would be impossible to generalize to target domains that include intents and slots that do not show up in the training set, since there will not be any parameter sharing among the classifiers. Instead of encoding slot names and intent names as discrete features in the target domain, we encode them using a continuous representation. To be specific, we learn a representation of slot names, slot values and intent names from their constituent words using semantic networks. We then check to see if these representations match the representations of utterances.

The architecture of our model is illustrated in Fig. 1. The model includes the following major components. (1) a word hashing layer to convert one-hot word representations into embedding vectors, (2) a bi-directional Long Short Term Memory networks(LSTMs) (Hochreiter & Schmidhuber, 1997) layer to capture features from word embeddings, (3) a intent specific bi-LSTMs layer combined with an semantic network to detect intent, (4) a slot specific bi-LSTMs layer powered with an attention component and semantic network to conduct slot filling. We first pretrain the model in a source domain with adequate annotated data. We then wish to allow the model to extend to new slot names, slot values and intent names in the target domain with little annotated data.

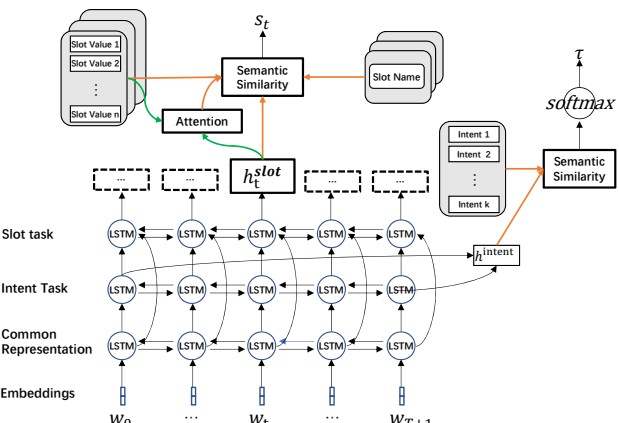

Figure 1: TSSM architecture for joint SLU. The orange lines are inputs for semantic similarity network and the green lines represent the inputs for attention network.

Let $\boldsymbol{w} = (w_0, w_1, w_2, \ldots, w_{T+1})$ represent the input word sequence with $w_0$ and $w_{T+1}$ being the beginning-of-sequence ($\langle bos \rangle$) and end-of-sequence ($\langle eos \rangle$) tokens, respectively. $T$ is the number of words. Let $\boldsymbol{s} = (s_0, s_1, \ldots, s_T)$ be the slot label sequence, where $s_0$ is a padded slot label that maps to the beginning-of-sequence token $\langle bos \rangle$ and $s_i \in \mathbb{S}$. Let $\boldsymbol{\tau} \in \mathbb{T}$ be the intent class. $\mathbb{S}$ and $\mathbb{T}$ are the slot and intent label space, respectively. Let $\boldsymbol{sn} = (\boldsymbol{sn}_1, \boldsymbol{sn}_2, \ldots, \boldsymbol{sn}_m)$ represent the $m$ slot names. $\boldsymbol{sn}_i = (w_1, \ldots, w_{m_i})$ is a vector of the words for the $i$-th slot name, where $m_i$ is its number of words. Each slot $s_i$ has a list of slot values $\boldsymbol{sv_i} = (\boldsymbol{sv}_{i,1}, \boldsymbol{sv}_{i,2}, \ldots, \boldsymbol{sv}_{i,n_i})$, where $n_i$ is the number of slot values for the $i$-th slot. A slot value is allowed to be a phrase like 'fried chicken' for slot 'Food'. Let $\boldsymbol{sv}_{i,j} = (w_1, \ldots, w_{n_{i,j}})$ be the words of the $j$-th slot value for $i$-th slot and $n_{i,j}$ is the number of words for this slot value.

## 3.1 BASE NETWORKS

We use a Pretrained DNN model as our baseline model (Goyal et al., 2018). The model has three layers of bi-LSTMs. The first layer is the common bi-LSTMs layer, which learns task related representations from input utterances. The learned representations are taken as input to the upper two bi-LSTMs layers, which are optimized separately for the slot-filling task and intent-detection task.

Our model differs from the baseline model in three aspects. First, Goyal et al. (2018) treated intents and slots as discrete symbols and train classifiers to make predictions on these symbols. Such an approach limits the knowledge transfer between two domains as the classifier layers (affine transform leading to the softmax) need be re-initialized when we transfer the model via fine-tuning in new domains, where the output labels are different. In our model, we encode intents and slots as continuous representations via their constituent words. The classification problems are transformed to semantic similarity problems and the whole model parameters could be transferred to new domains.

The second difference is the usage of *gazetteers* (lists of slot values for a slot). Goyal et al. (2018) used gazetteer features as an additional input. Such features are binary indicators of the presence of an n-gram in a gazetteer. In our model, we used an attention network (Section 3.3) to encode external slot values into a semantic vector that represents the slot from the value perspective, which suits our semantic framework naturally.

Finally, there are no connections between the upper two layers in the baseline by Goyal et al. (2018). However, we believe that the output of intent specific bi-LSTMs layer could benefit the slot detection task. As one of our contributions, in our network, we concatenate the output of common bi-LSTMs layer and intent bi-LSTMs layer to feed to the slot bi-LSTMs layer.

For the intent specific bi-LSTMs layer, we concatenate the last hidden state of each direction to acquire the global representation of the whole utterance $\mathbf{h}^{intent} = [\overrightarrow{\mathbf{h}}_{T+1}^{intent}, \overleftarrow{\mathbf{h}}_0^{intent}]$. Such an operation converts utterances with variable lengths into a fixed-length vector, with which the information throughout the entire utterance can be captured. We have also tried other methods like take Mean-Pooling or Max-Pooling of bi-LSTM hidden states to acquire $\mathbf{h}^{intent}$ but get similar results.

## 3.2 SEMANTIC NETWORKS

We wish to build a representation-learning mechanism for the SLU task that can generalize to unseen words and labels. For slot names, slot values and intent names, we use the same semantic network and transform them to fixed-length representation vectors. Take slot name $\boldsymbol{sn}_i = (w_1, w_2, \ldots, w_{m_i})$ as an example, the semantic network first extracts the word embedding $E(w_i)$ for each word. $E(w_i)$ is then fed to a feed forward layer to get the non-linear semantic features. The last is a Mean-Pooling layer, which applies the mean operations over each dimension of semantic features across all words. The output is treated as the semantic representation for $\boldsymbol{sn}_i$,

$$\mathbf{r}_i^{slot\_name} = \frac{1}{m_i} \sum_{j=1}^{m_i} relu\left(E(w_j)^\intercal \mathbf{W}_{slot\_transfrom} + b_{slot\_transfrom}\right) \tag{1}$$

where $\mathbf{W}_{slot\_transfrom}$ and $b_{slot\_transfrom}$ are the learned linear projection matrix and bias for slot name semantic network respectively. In the same way, we could get the semantic representation $\mathbf{r}_{i,j}^{slot\_value}$ for slot value $\boldsymbol{sv}_{i,j}$ and $\mathbf{r}_i^{intent\_name}$ for the intent name $\tau_i$. More sophisticated vector-space semantic representations of the slots/intents are an area for future work, but we believe the simple Mean-Pooling would be a proper choice here since intent names, slot names and slot values are typically very short and composed by entity names.

## 3.3 SLOT VALUE ATTENTION NETWORKS

With semantic networks, we formulate a semantic representation $\mathbf{r}_{i,j}^{slot\_value}$ of each slot value $\boldsymbol{sv}_{i,j}$ from the $i$-th candidate slot $s_i$. The whole slot value representation list is $\left(\mathbf{r}_{i,1}^{slot\_value}, \ldots, \mathbf{r}_{i,n_i}^{slot\_value}\right)$. At each decoding step $t$, the slot bi-LSTMs layer output state $\mathbf{h}_t^{slot}$ is connected to each slot's slot values' representations to calculate the similarity scores. Take slot $s_i$ as an example, we compute an attention weight $\alpha_{t,i,j}$ for slot value $\boldsymbol{sv}_{i,j}$ via a bi-linear operator, which reflects how relevant or important slot value $\boldsymbol{sv}_{i,j}$ is to the current slot hidden states $\mathbf{h}_t^{slot}$,

$$\alpha_{t,i,j} \propto \exp((\mathbf{r}_{i,j}^{slot\_value})^\intercal \mathbf{W}_v \mathbf{h}_t^{slot}) \tag{2}$$

where $\mathbf{W}_v$ is a parameter matrix. The overall slot value representation for slot $s_i$ would be

$$\mathbf{r}_{t,i}^{slot\_values} = \sum_{j}^{n_i} \alpha_{t,i,j} \mathbf{r}_{i,j}^{slot\_value}. \tag{3}$$

### 3.4 TRAINING PROCEDURE

The source domain data containing utterances and associated intents/slots is used to train the model. The idea behind this model is to learn the representation for both utterances and intents/slots such that utterances with the same intents/slots are close to each other in the continuous semantic space. We define a semantic score between an utterance and an intent/slot using the similarity between their representation embeddings $\mathbf{h}$ and $\mathbf{r}$ by $Sim(\mathbf{h}, \mathbf{r}) = \mathbf{h}^\mathsf{T} \mathbf{W}_s \mathbf{r}$. $\mathbf{W}_s$ is the corresponding similarity matrix for slot or intent task.

The posterior probability of a possible intent given an utterance is computed based on the semantic score through a softmax function

$$P(\tau = \tau_i | \mathbf{h}^{intent}, \boldsymbol{w}) = \frac{\exp\left(Sim(\mathbf{h}^{intent}, \mathbf{r}_i^{intent\_name})\right)}{\sum_{\tau_j in \mathbb{T}} \exp\left(Sim(\mathbf{h}^{intent}, \mathbf{r}_j^{intent\_name})\right)} \tag{4}$$

where $\tau_i$ is an intent candidate. Similarly, we can get the posterior probability of a possible slot at each time. Since we have the representations of both slot name and slot value, we first add the semantic scores from slot name and slot value and then feed to the softmax function to get the overall slot posterior probability

$$Sim(s_i, \mathbf{h}_t^{slot}) = Sim(\mathbf{h}_t^{slot}, \mathbf{r}_i^{slot\_name}) + Sim(\mathbf{h}_t^{slot}, \mathbf{r}_i^{slot\_values}) \tag{5}$$

$$P(s_t = s_i | \mathbf{h}_t^{slot}, \boldsymbol{w}) = \frac{\exp\left(Sim(s_i, \mathbf{h}_t^{slot})\right)}{\sum_{s_j \in \mathbb{S}} \exp\left(Sim(s_j, \mathbf{h}_t^{slot})\right)} \tag{6}$$

where $s_i$ is an slot candidate. For model training, we maximize the likelihood of the correctly associated intents/intents given all training utterances. The parameters $\mathbf{W}$ of the model are optimized by an joint negative log likelihood objective:

$$\mathcal{L} = -\sum_{k=1}^{l} \log P\left(\tau = \tau_i | \mathbf{h}_{\mathbf{w}_k}^{intent}, \boldsymbol{w}_k\right) - \sum_{k=1}^{l} \sum_{t=0}^{T_k} \log P\left(s_t = s_i | \mathbf{h}_{t,\mathbf{w}_k}^{slot}, \boldsymbol{w}_k\right) \tag{7}$$

where $l$ is the total number of training utterances, $\mathbf{w}_k$ is the $k$-th input utterance. The overall objective function combines the intent classification task and slot filling task objectives. We observe that this multitask architecture achieves better results than separately training intent and slot models. The model is optimized using mini-batch stochastic gradient descent(SGD) (Huang et al., 2013).

### 3.5 TRANSFER TO A NEW DOMAIN

We train our TSSM using labeled data from source domain $D_s$. This stage is called pretraining, where the networks learn to extract semantic relationships between utterance representations and intent/slot representations. The model is able to adapt to new domains, since the seen and unseen intents/slots representations are in the same semantic space, use the same shared composition semantic network with the same unsupervised word embeddings as input.

After the pretraining, we fine-tune our model on the target domain $D_t$ with less annotated data and with unseen intents/slots. The model for the new domain $D_t$ are the same as that for $D_s$ and are initialized by the pretrained network parameters. We tried fine-tune different components of the model. We fix some parts and fine-tune on the other parts. After experiments on several combinations of components, we found that fine-tune the whole model gives us the best performance.

For zero-shot learning where no fine-tune data is available, in order to predict new intents/slots on new utterances, the TSSM calculates semantic similarities. We transform each input utterance $\mathbf{w}$ into a vector $\mathbf{h}^{intent}$ using the base network, and then estimate its semantic similarity with all intents including seen and unseen intents. The vector representations for new intents can be generated from the trained semantic network by feeding the word embedding vectors of new intents as the input. For the utterance $\mathbf{w}$, the estimated semantic score of the $i$-th intent is defined as $Sim(\mathbf{h}^{intent}, \mathbf{r}_i^{intent})$ in Equation 3.4. Then predicted intent for each utterance is decided according to the estimated semantic scores. Predicting slot sequences is similar to predicting intents except that the calculation is conducted step by step at each position.

## 4 EXPERIMENTS AND RESULTS

In this experiment, we use five popular public dialogue datasets from five domains and three datasets obtained from practical applications. The CamHotel dialogue dataset (Hotel) (Wen et al., 2015) records some dialogues about booking a hotel room in Cambridge. The CamRestaurant dialogue dataset (Restaurant) (Wen et al., 2016c;a) contains dialogue records about booking a restaurant in Cambridge. The Laptop and TV dialogue dataset (Wen et al., 2016b) are dialogues about buying Laptops and TVs. The Airline Travel Information System dataset (Atis) (Price, 1990) contains dialogues about airline travel, such as the flight time, the destination, etc. Weilidai, Huobijijin and Weizhongyouzhe are three financial products from WeBank, an internet bank in China. While Weilidai is very popular and its online customer service accumulated a lot of queries, Huobijijin and Weizhongyouzhe are newly developed products and have only few queries. We would like to pretrain on Weilidai queries and fine-tune on Huobijijin and Weizhongyouzhe queries to improve the online customer service's experience of the last two products.

In order to thoroughly evaluate the transfer learning ability on more diversified domains, we make five different pairs of source and target domains, as shown in table 1. In case when we have multiple source domains, all source domains will be combined into a large source domain. For each pair of the source and target domain pair, we first pretrain the model on the source domain. Then we fine-tune the model on the target domain training data, finally we evaluate the fine-tuned model on the target domain test set. We use accuracy to evaluate the intent classification task and we use accuracy and F1 score to evaluate the slot filling task. Apart from the Pretrained DNN (Goyal et al., 2018), we also compare TSSM to Max Entropy model and Conditional Random Field model (CRF).

Table 1: Data Sizes of the source and target domains.

| Source Domain | Target Domain | # Source Sentence | # Target Sentence |
|---|---|---|---|
| Restaurant, Laptop, TV, Atis | Hotel | 48566 | 2168 |
| Restaurant, Hotel, Laptop, TV | Atis | 45756 | 4978 |
| Atis | Hotel | 4978 | 2168 |
| Weilidai | Huobijijin | 5730 | 927 |
| Weilidai | Weizhongyouzhe | 5730 | 1053 |

### 4.1 RESULTS UNDER DIFFERENT SIZES IN TARGET-DOMAIN DATA SETS

In order to evaluate the transfer learning performance of the proposed model, we compare the proposed model with the baselines when we have different number of target domain training data. After pretraining in the source domain, we fine-tune the model on different percentages of the target domain data, and we evaluate the performance of the models on the test set. We use $50\%$ of the target domain data as test set, and we vary the percentage of train set used from $5\%$ to $46\%$.

The results are shown in Figure 2. As a higher percentage of target domain data is used in fine-tuning, the performances of all models improve. Generally, transfer learning models perform better than non-transfer models, which implies that transfer learning can help improve the performance in new domain. The proposed TSSM model outperforms the baseline model significantly in almost all cases. In some cases, the proposed TSSM is not better than DNN. This might because our model has more parameters than the baseline DNN model and in some cases simper model performs better.

Compared with intent classification task, the slot filling task benefit more from transferring from the source domain. This is because the intent has larger domain differences compared with the slots. For example, the intent in the hotel domain contains verbs such as "inform" "inform_no_match" while the intent in the Atis domain contains nouns such as "flight" "aircraft". While the slots in all domains mostly contain nouns. The improvement of transferring from Atis to Hotel dataset is worse than the other source and target pairs. We think this is because the intent and slot difference between "Atis" and 'Hotel' domains are bigger than the domains difference of other tasks.

### 4.2 RESULTS UNDER DIFFERENT NUMBERS OF NEW LABELS IN THE TARGET DOMAIN

In order to evaluate the generalization ability of the TSSM model, we test the model on previously unseen labels. More specifically, we remove zero, one and two labels and their corresponding training instances from the target domain training set. Then we evaluate TSSM on the target domain

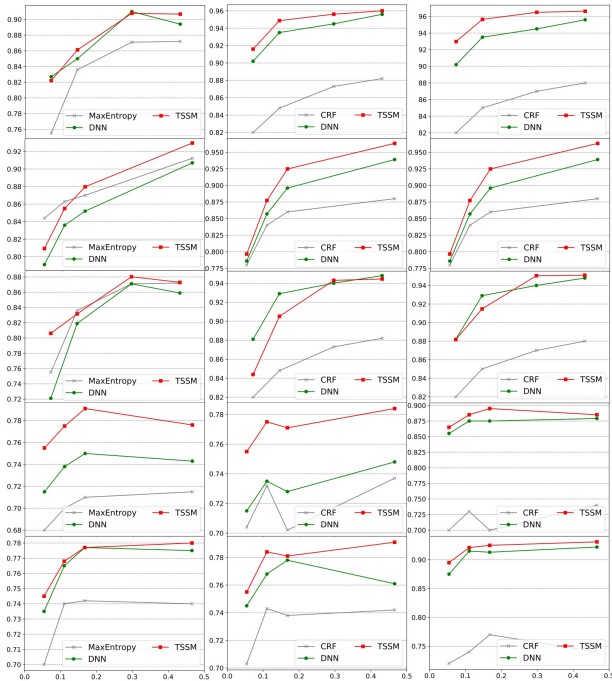

Figure 2: Performance comparison under different percentage of training data used in the target domain fine-tuning. In each figure, X-axis is the percentage of data used in the target domain, y-axis is the performance (higher is better). The three columns corresponds to intent accuracy (left column), slot accuracy (middle column) and slot f1 score (right column). The five rows corresponds to different source and target domain pairs. From the top to the bottom are "Restaurant, Laptop, TV, Atis" to "Hotel" "Restaurant, Hotel, Laptop, TV" to "Atis" "Atis" to "Hotel" "Weilidai" to "Weizhongyouzhe" and "Weilidai" to "Huobijijin".

test set which has full lists of labels. The results are shown in Figure 3. Here we use $10\%$ of all target domain data as the training set, when the percentage of target domain training data increases to $20\%$ or $30\%$, the results are similar. As the number of unseen label increases in the target domain test set, the performance of all model drops significantly. Transfer learning methods again out-performs none-transfer learning baselines, which demonstrate that transferring knowledge from source domain is generally beneficial to the target domain.

As the number of unseen label increases, the performance drop of the TSSM model is significantly slower than the other baselines models. Compared with a softmax classification layer, the semantic similarity module works better since it can still compute semantic similarity for unseen labels. With a pretrained word embeddings, the TSSM can theoretically compute semantic similarity for arbitrary label as long as the label has a word embedding. The experiments demonstrate that the TSSM model can effectively deal with cold start labels that is not seen in the target domain training data.

### 4.3 ZERO-SHOT TRANSFER LEARNING

In the extreme case, there could be no training instances in the target domain at all. In order to test the proposed TSSM model on such extreme case, we report the performances of the TSSM model when there is no target domain training instances. The result is shown in Table 2. The random guess performance is $0.05$ for intent classification accuracy, $0.02$ for slot accuracy and $0.02$ for slot f1. The TSSM have an performance much better than random guess. In such case, both the Pretrained DNN, the Max Entropy model and the CRF model could not work, since all of them require target domain training instances to initialize their softmax layer.

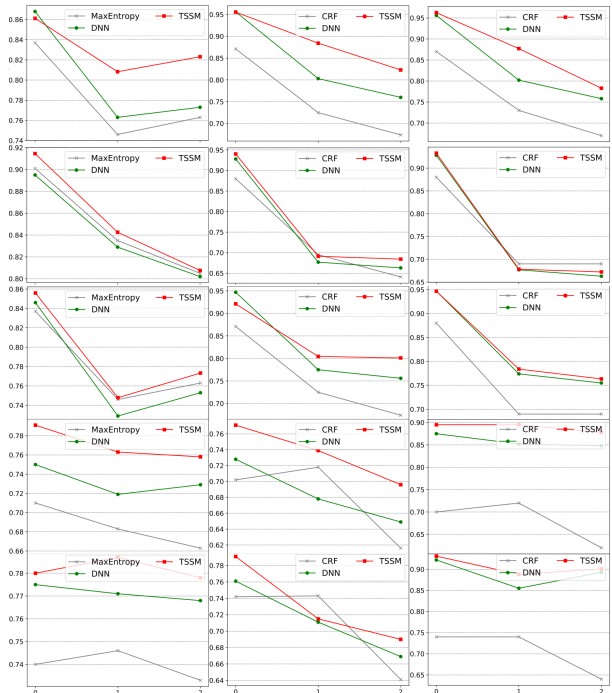

Figure 3: Performance vs the number of unseen intent/slot in the target domain test set, where 10% of the total target domain data is used. In each figure, X-axis is the number of unseen intent/slots in the target domain test set, y-axis is the performance (higher is better).

Table 2: The zeros-shot performance of TSSM

| Source and Target | Intent Accuracy | Slot Accuracy | Slot F1 |
|---|---|---|---|
| nohotel2hotel | 0.787 | 0.871 | 0.870 |
| atis2hotel | 0.478 | 0.756 | 0.759 |
| noatis2atis | 0.220 | 0.617 | 0.616 |

## 4.4 ABLATION EXPERIMENT

One of the improvements we have made is to use the intent bi-LSTMs output states as input features to the slot bi-LSTMs, we call this mechanism "intent2slot". To evaluate the effectiveness of "intent2slot" mechanism, we compare the TSSM with its ablation variant without "intent2slot", and the result is shown in Table 3. The results demonstrate that the "intent2slot" mechanism helps to improve intent classification performance. The parameters in intent bi-LSTMs layer might have obtained additional supervision from the upper slot filling task. The improvements for transferring from Restaurant, Laptop, TV, Atis to Hotel is not obvious and the reason needs further experiments.

Table 3: The TSSM with/without "intent2slot"

| Source and Target | Intent Accuracy | | Slot Accuracy | | Slot F1 | |
|---|---|---|---|---|---|---|
| | TSSM-abl | TSSM | TSSM-abl | TSSM | TSSM-abl | TSSM |
| nohotel2hotel | 0.949 | **0.952** | 0.924 | 0.926 | 0.938 | 0.935 |
| atis2hotel | 0.945 | **0.952** | 0.937 | 0.938 | 0.945 | 0.945 |
| noatis2atis | 0.807 | **0.840** | 0.891 | 0.896 | 0.899 | 0.888 |

## 5 CONCLUSIONS

In this paper, we introduced a deep semantic similarity network to transfer from domains with sufficient labeled data to low resource domains. The experiments illustrated that our proposed methods outperform the state-of-the-art baselines in several experimental settings. In the future, we will consider different regularization choices to make our model more robust in changing data situations. We will also consider different data set types for dialog systems.

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
