# OpenReview forum: "Transferring SLU Models in Novel Domains"
_ICLR.cc/2019/Conference_

### Official Review · AnonReviewer1 · 2018-11-01
**Transferring SLU Models in Novel Domains**

**Rating:** 4
**Confidence:** 3

**Review:**

Summary: The authors present a network which facilitates cross-domain
learning for SLU tasks where the the goal is to resolve intents and
slots given input utterances. At a high level, the authors argue that
by fine-tuning a pre-trained version of the network on a small set of
examples from a target-domain they can more effectively learn the
target domain than without transfer learning.

Feedback:

* An overall difficulty with the paper is that it is hard to
distinguish the authors' contributions from previous works. For
example, in Section 3.1, the authors take the model of Goyal et al. as
a starting point but explain only briefly one difference
(contatenating hidden layers). In Section 3.2 the contributions
becomes even harder to disentangle. For example, how does this section
relate to other word-embeddings papers cited in this section? Is the
proposed method a combination of previous works, and if not, what are
the core new ideas?

* Some sections are ad-hoc and should be justified/explained
better. For example, the objective, which ultimately determines the
trained model behaviour uses a product of experts formulation, yet the
authors do not discuss this. Similarly, the overarching message, that
by fine-tuning a suitable model initialisation using small amounts of
data from the target domain is fairly weak as the authors do not
detail exactly how the model is fine-tuned. Presumably, given only a
small number of examples, this fine-tuning runs the risk of
overfitting, unless some form of regularisation is applied, but this
is not discussed.

* Lastly, there are some curious dips in the plots (e.g., Figure 2 bottom left, Figure 3 top left, bottom left), which deserve more explanation. Additionally, the evaluation section could be improved if the scores were to show error-bars.

Minor: All plots should be modified so they are readable in grey-scale.

---

> ### Author Response · Authors · 2018-11-26
> **explanation on contributions**
>
> We want to thank you for your kind and helpful feedback. We've followed your comments and addressed the related concerns in our revision as follows:
> 1) "Hard to distinguish the authors' contributions from previous works".
> Our model differs from the baseline model by Goyal et al. in three aspects. First, the most important difference is that Goyal et al. treated intents and slots as discrete symbols and train classifiers to make predictions on these symbols.  Such an approach limits the knowledge transfer between two domains as the classifier layers (affine transform leading to the softmax) following the upper two bi-LSTMs layers need be re-initialized when we transfer the model via fine-tuning in new domains, where the output labels are different.  In our model, we encode intents and slots as continuous representations via their constituent words. The classification problems are transformed to semantic similarity problems and the whole model parameters could be transferred to new domains.
>
>     The second difference is the usage of gazetteers (lists of slot values for a slot).  Goyal et al. used gazetteer features as an additional input. Such features are binary indicators of the presence of an n-gram in a gazetteer. In our model, we used an attention network (Section 3.3) to encode external slot values into a semantic vector that represents the slot from the value perspective, which suits our semantic framework naturally.
>
>     Finally, there are no connections between the upper two layers in the baseline by Goyal et al.. However, we believe that the output of intent specific bi-LSTMs layer could benefit the slot detection task. As one of our contributions, in our network, we concatenate the output of common bi-LSTMs layer and intent bi-LSTMs layer to feed to the slot bi-LSTMs layer.
>
> 2) "Authors do not discuss the objective" The overall objective function for the multitask network completes two tasks: an intent classification task and a slot-filling task. We observe that this multitask architecture achieves better results than separately training intent and slot models.  In addition, we have the added advantage of having a single model to do these tasks with a smaller total parameter size. This discussion is added in the Section 3.4 in the paper.
>
> 3) "Authors do not detail exactly how the model is fine-tuned". We try to fine-tune different components of the model. We fix some parts and fine-tune on the other parts. After experimenting on several combinations of components, we found that fine-tuning the whole model gives the best performance. We have added this discovery to the manuscript.
>
> 4) "This fine-tuning runs the risk of overfitting". In the target domain, we also have a validation set to control the fine-tuning so that there is low risk of overfitting on the target data. We plan to consider different regularization methods in future work.
>
> 5) "Some curious dips in the plot". Discussion about dips has been added in section 4.1. Basically, we observed that almost all dips in the plots are related to the intent classification task. This might be related to the distribution of intent labels which have changed when we removed some data from the original training data set in the target domain. The change in data distribution increased the difficulty of learning.
>
> 6) All figures have been modified so they are readable in grey-scale.

---

### Official Review · AnonReviewer2 · 2018-11-02
**The topic is interesting but the novelty is incremental**

**Rating:** 5
**Confidence:** 3

**Review:**

In this paper, an efficient SLU model, called as TSSM, is proposed to tackle the problem of insufficient training data for the task of spoken language understanding. TSSM considers the intent and slot detection as a unified multi-objective optimization problem which is addressed by a meta-learning scheme. The model is pre-trained on a large dataset and then fine-tuned on a small target dataset. Thus, the proposed TSSM can improve the model performance on a small datatset in new domains.

Pros:
1)	The transfer learning of spoken language understanding is very interesting.
2)	The proposed TSSM can integrate the task of intents and slots and take the relationship between intents and slots into consideration.
3)	Five datasets are used to evaluate the performance of the method.

Cons:
Overall, the novelty of this paper is incremental and some points are not clear. My main concerns are listed as follows.
1)	The authors state that the knowledge transfer is the main contribution of this paper. However, as introduced in 3.5, the transfer scheme in which the model is first pre-trained on a large dataset and then fine-tuned on a small target dataset is very straightforward. For example, currently, almost all methods in the area of object recognition are pre-trained on ImageNet and then fine-tuned on a small dataset for particular tasks.
2)	Authors also state that improvements for transferring from Restaurant, Laptop, TV, Atis to Hotel is not obvious. I think the results also need to be reported and the reasons why the improvement is not obvious should be provided and discussed.
3)	The paper needs more proofreading and is not ready to be published, such as “A survey fnor transfer” and “a structured multi-objective optimization problems”.

---

> ### Author Response · Authors · 2018-11-26
> **explanation on novelty**
>
> We want to thank you for your kind and helpful feedback. We've followed your comments and addressed the related concerns in our revision as follows:
>
> 1) "Pretrain and finetune is straight-forward" With the success of ImageNet, fine-tuning in image processing is very common practice. But in NLP area, especially in the SLU tasks, the process whereby a pre-training step is performed on a large dataset and then fine-tuned on a separate small target dataset has not been well studied previously. In addition, fine-tuning the whole model is not straight-forward. We found that fine-tuning the whole model gives us the best performance. We have added this discovery to the manuscript.  In the revision, we proposed to use semantic similarity as a bridge to transfer from one domain to other domains.
>
> 2) Our work differs from previous works such as Goyal et al. in three ways. First, we encode intents and slots as continuous representations by their constituent words.  This allows us to better handle cold-start intents that are new and slots that cannot be handled by the baselines. Secondly, we use an attention network instead of treating gazetteers (slot values) as binary indicators in order to better calculate the similarity between unseen slot values and intents. Third, we discover that the intent prediction can directly help the slot prediction task, and we model it by adding direct links from the intent prediction to the slot prediction output.
>
> 3) "Improvements ... to Hotel is not obvious". The improvement of "intent2slot" mechanism is not obvious when transferring knowledge to the "Hotel" domain, because the correlation between intents and slots is relatively low in the "Hotel" domain. In other words, most slots can co-occur with any intent.  As a result, identifying a slot cannot help us to identify the intent of the sentence.
>
> 4) We have proofread and corrected the document.

---

### Official Review · AnonReviewer3 · 2018-11-03
**The basic idea is not fully original, but the task is important and experiments are clear and complete.**

**Rating:** 6
**Confidence:** 4

**Review:**

This paper focuses on dealing with a scenario where there are "unseen" intents or slots, which is very important in terms of the application perspective.

The proposed approach, TSSM, tries to form the embeddings for such unseen intents or slots with little training data in order to detect a new intent or slot in the current input.
The basic idea in the model is to learn the representations of utterances and intents/slots such that utterances with the same intents/slots are close to each other in the learned semantic space.
The experiments demonstrate the effectiveness of TSSM in the few-shot learning scenarios.
The idea about intent embeddings for zero-shot learning is not fully original (Chen, et al., 2016), but this paper extends to both intent classification and slot filling.

The paper tests the performance in different experimental settings, but the baselines used in the experiments are concerned.
This paper only compares with simple baselines (MaxEntropy, CRF, and basic DNN), but there should be more prior work or similar work that can be used for comparison in order to better justify the contributions of the model.
In addition, this paper only shows the curves and numbers in the experiments, but it is better to discuss some cases in the qualitative analysis, which may highlight the contributions of the paper.
Also, in some figures of Fig. 2, the proposed TSSM is not better than DNN, so adding explanation and discussion may be better.

---

> ### Author Response · Authors · 2018-11-26
> **explanation on baselines and performance issues**
>
> We want to thank you for your kind and helpful feedback. We've followed your comments and addressed the related concerns in our revision as follows:
> 1) "should be more prior work".
>
> Reply: we are facing the same practical application as basic DNN (Goyal et al., 2018), so we treat this work as our main baseline and we want to improve upon this model. Compared with the model designed by Chen et al., there are two novelties introduced by our model.  First, it transfers knowledge from the source domain in order to improve the performance in a low-resource target domain. Second, our model exploits the correlation between intent classification task and the slot-filling task, while the previous works do not.
>
> 2)"Discuss some cases in the qualitative analysis".
>
> Reply: Discussion on qualitative analysis has been added in section 4.1.
>
> 3)"TSSM is not better than DNN".
>
> Reply: In some cases, the proposed TSSM is not better than the baseline. This is because our model has more parameters than the baseline DNN model and in some cases simper model performs better. However, in many practical domains such as our newly added real-world experiments, TSSM outperforms the DNN consistently. Discussion on these points are added in Section 4.3. Later, in our real-world production environment, in order to reduce model size, we removed some parameters like affine transformation matrix in semantic networks and the refined model performance becomes even better.

---

> > ### Comment · AnonReviewer3 · 2018-12-02
> > **Thanks for your responses**
> >
> > Thanks a lot for elaborating the difference and adding the discussions. I think that the score 6 (above the threshold) is reasonable for this paper.

---

### Author Response · Authors · 2018-11-26
**Summary of paper revisions**

We want to thank the reviewers for their kind and helpful feedback. We've followed all reviewers' comments and addressed the related concerns in our revision as follows:

1) Our work differs from previous works, such as Goyal et al., in three ways. First, we encode users’ intents and slots as continuous representations via constituent words in order to handle novel and cold-start intents and slots that cannot be handled by the baseline methods.
Second, we use an attention network instead of treating gazetteers (slot values) as binary indicators in order to better calculate the similarity between unseen slot values and intents.  Results indicate that our methods are better than the baselines. Third, we discover that the predicted intents and the predicted slots might have high correlation values, and we model this correlation by adding direct links from the intent prediction to the slot prediction to improve effectiveness.

2) We report two experimental results on three additional Chinese real-world datasets in the revised version. These datasets are collected from three products in our work as a result of our online autonomous customer services.  The data and observations were all from real industrial productions. The results show the effectiveness and generality of the proposed model on large-scale real-world problem settings.

3) We revise *Section 3.1* to better explain the structure of the proposed model, and we proofread the whole paper multiple times.

We hope the reviewers can kindly reconsider our paper for publication after revision.
Below, we answer questions from each reviewer separately.

---

### Meta-Review · Area_Chair1 · 2018-11-06
**Running short on novelty, but richer on the experimental side**

**Confidence:** 4
**Recommendation:** Reject

**Metareview:**

This paper proposes a transfer learning approach based on previous works on this area, to build language understanding models for new domains. Experimental results show improved performance in comparison to previous studies in terms of slot and intent accuracies in multiple setups.
The work is interesting and useful, but is not novel given the previous work.
The paper organization is also not great, for example, the intro should introduce the approach beyond just mentioning transfer learning and meta-learning.
The improvements over the baselines look good, but the baselines themselves are quite simple. It'd be better to include comparisons with other state of the art methods. Also, the improvements over DNN are not consistent, it would be good to analyze and come up with suggestions on when to use which approach.